# Hispidulin Inhibits the Vascular Inflammation Triggered by *Porphyromonas gingivalis* Lipopolysaccharide

**DOI:** 10.3390/molecules28186717

**Published:** 2023-09-20

**Authors:** Yeon Kim, Hoyong Lee, Hyun-Joo Park, Mi-Kyoung Kim, Yong-Il Kim, Hyung Joon Kim, Soo-Kyung Bae, Yung-Jin Kim, Moon-Kyoung Bae

**Affiliations:** 1Department of Oral Physiology, School of Dentistry, Pusan National University, Yangsan 50612, Republic of Korea; 2Periodontal Disease Signaling Network Research Center (MRC), Pusan National University, Yangsan 50612, Republic of Korea; 3Dental and Life Science Institute, Pusan National University, Yangsan 50612, Republic of Korea; 4Department of Molecular Biology, Pusan National University, Busan 46241, Republic of Korea; 5Department of Orthodontics, School of Dentistry, Pusan National University, Yangsan 50612, Republic of Korea; 6Department of Dental Pharmacology, School of Dentistry, Pusan National University, Yangsan 50612, Republic of Korea

**Keywords:** hispidulin, *Porphyromonas gingivalis*, lipopolysaccharide, monocyte, vascular endothelial cells, vascular inflammation

## Abstract

Hispidulin is a natural bioactive flavonoid that has been studied for its potential therapeutic properties, including its anti-inflammatory, antioxidant, and neuroprotective effects. The aim of this study was to explore whether hispidulin could inhibit the endothelial inflammation triggered by *Porphyromonas gingivalis* (*P. gingivalis*) lipopolysaccharide (LPS). The adhesion of monocytes to the vascular endothelium was evaluated through in vitro and ex vivo monocyte adhesion assays. We analyzed the migration of monocytes across the endothelial layer using a transmigration assay. The results showed that treatment with hispidulin decreased the *P. gingivalis* LPS-induced adhesion of monocytes to endothelial cells and their migration by suppressing the *P. gingivalis* LPS-triggered expression of intercellular adhesion molecule-1 (ICAM-1) through downregulating nuclear factor-қB (NF-қB). In addition, hispidulin inhibited *P. gingivalis* LPS-induced mitogen-activated protein kinases (MAPKs) and AKT in endothelial cells. Altogether, the results indicate that hispidulin suppresses the vascular inflammation induced by *P. gingivalis* LPS. Mechanistically, it prevents the adhesion of monocytes to the vascular endothelium and migration and inhibits NF-қB, MAPKs, and AKT signaling in endothelial cells.

## 1. Introduction

Periodontal disease is a frequent inflammatory disorder within the oral cavity affecting the teeth and tooth-supporting tissues, and it is caused primarily by *P. gingivalis*, a major Gram-negative anaerobic periodontal pathogen [1,2]. Growing evidence indicates that periodontal disease has an adverse impact on several systemic diseases, including diabetes, Alzheimer’s, colorectal cancer, and atherosclerosis [3]. *P. gingivalis* and its LPS can cause periodontitis and directly lead to systemic inflammation by invading the bloodstream, which may trigger or exacerbate vascular inflammatory processes, especially atherosclerosis [4]. Following an infection with *P. gingivalis,* leukocytes attach to and penetrate the endothelial layer within inflamed periodontal tissue during the vascular inflammatory process [5]. This process is dependent on the presence of specific cell adhesion molecules on the surface of the vascular endothelium, namely VCAM-1 and ICAM-1 [6].

Flavonoids are plant-derived polyphenolic compounds with various beneficial effects in relieving the initiation and progression of periodontal disease [7]. The administration of epigallocatechin-3-gallate, the major flavonoid in green tea, to apolipoprotein E (ApoE)-knockout mice injected with *P. gingivalis* decreased atherosclerotic plaque formation, the serum levels of pro-inflammatory cytokines, and atherosclerotic risk factors [8]. Hispidulin (4′,5,7-trihydroxy-6-methoxyflavone) is a natural phenolic flavonoid present in various plants, including *Saussurea involucrata*, *Arrabidaea chica*, *Salvia involucrata*, and *Grindelia argentina*, and it has various pharmacological activities, including antioxidant, anticancer, antifungal, anti-osteoporotic, neuroprotective, and anti-inflammatory activities [9,10,11,12]. In recent years, some studies have investigated the anti-inflammatory effects of hispidulin on various pro-inflammatory stimuli in different cell types and tissues. Hispidulin alleviates skin, airway, and allergic inflammation by downregulating the production of pro-inflammatory cytokines and chemokines [13,14,15]. Additionally, hispidulin suppresses neuroinflammatory responses in activated microglia via the reduction of inflammatory mediators [16]. More recently, hispidulin has been shown to block the angiogenic properties of endothelial cells, such as vascular endothelial growth factor-induced proliferation, migration, and tubular formation [17]. However, little attention has been given to the inhibitory effect of hispidulin on vascular inflammation. 

The aim of this study was to determine whether hispidulin prevented *P. gingivalis* LPS-induced endothelial inflammation and to propose possible anti-inflammatory mechanisms of hispidulin in endothelial cells.

## 2. Results

### 2.1. Hispidulin Decreases the P. gingivalis LPS-Induced Expression of Inflammatory Cytokines in Vascular Endothelial Cells

To determine the optimal concentration of hispidulin for in vitro experiments, we assessed the viability of human umbilical vein endothelial cells (HUVECs) incubated with different concentrations of hispidulin (1, 5, 10, or 20 μM) for 1–3 days using the methylthiazolyl tetrazolium (MTT) assay. No cytotoxicity was observed when cells were treated with 1 μM hispidulin. At 5 μM or 10 μM, hispidulin slightly decreased the relative viability over 3 days. After 2–3 days of treatment with 20 μM hispidulin, the proliferation of HUVECs was significantly reduced (approximately 20%) compared to that in the control group (Figure 1a). During vascular inflammation, pro-inflammatory cytokines, such as monocyte chemoattractant protein-1 (MCP-1), Interleukin-1 β (IL-1β), and Interleukin-8 (IL-8), play critical roles in the recruitment of monocytes to endothelial cells at inflammation sites [18]. Hispidulin treatment decreased the mRNA levels of MCP-1, IL-1β, and IL-8 upregulated by *P. gingivalis* LPS (Figure 1b).

### 2.2. Hispidulin Decreases the P. gingivalis LPS-Induced Expression of ICAM-1 in Endothelial Cells 

Vascular endothelial cells express cell adhesion molecules such as ICAM-1 in response to injury or inflammation, thereby allowing for the adhesion of leukocytes to the endothelium [19]. To determine the potential role of hispidulin in endothelial dysfunction, we examined its effect on *P. gingivalis* LPS-induced ICAM-1. *P. gingivalis* LPS significantly increased ICAM-1 mRNA levels in HUVECs; however, treatment with hispidulin reduced their expression (Figure 2a). As shown in Figure 2b, a Western blot showed that hispidulin decreases the ICAM-1 protein levels induced by *P. gingivalis* LPS. In addition, the surface expression of the ICAM-1 protein was evaluated using flow cytometry, a standard technique for assessing cell-surface-bound proteins. Hispidulin reduced the *P. gingivalis* LPS-induced cell surface expression of the ICAM-1 protein (Figure 2c).

### 2.3. Hispidulin Inhibits Monocyte Adhesion to P. gingivalis LPS-Stimulated Vascular Endothelial Cells 

Next, we investigated whether hispidulin affected the adhesion of monocytes to *P. gingivalis* LPS-stimulated endothelial cells, which is a critical step in vascular inflammation [20]. HUVECs were exposed to *P. gingivalis* LPS or *P. gingivalis* LPS in the presence of hispidulin and co-cultured with fluorescence-labeled THP-1 monocytes. *P. gingivalis* LPS stimulated the adhesion of THP-1 cells to HUVECs; however, this effect decreased significantly upon the treatment of HUVECs with hispidulin (Figure 3a). Next, we conducted an ex vivo endothelial-monocyte adhesion assay using fluorescently labeled THP-1 cells and the aorta isolated from a Sprague Dawley rat. The number of fluorescently labeled monocytes adhering to the aortic endothelium increased significantly following *P. gingivalis* LPS treatment relative to the untreated control (Figure 3b); however, it dropped significantly upon hispidulin addition.

### 2.4. Hispidulin Decreases the P. gingivalis LPS-Induced Transendothelial Migration of Monocytes

Once monocytes adhere to the endothelium, they transmigrate through the endothelial layer as a subsequent progressive step in atherosclerotic lesion formation [21,22]. We used a transmigration assay to assess whether the migration of *P. gingivalis* LPS-stimulated THP-1 cells through the endothelial cell layer was modulated by hispidulin treatment. HUVECs were seeded in transwell inserts to reach a confluence. Subsequently, fluorescently labeled THP-1 cells were introduced into the upper chamber and permitted to transmigrate for 24 h. The number of THP-1 cells that migrated via the endothelial layer into the lower chamber was counted. We observed that *P. gingivalis* LPS significantly increased (by 7-fold) the transmigration of THP-1 cells across HUVEC monolayers relative to control cells (Figure 3c); however, it was markedly retarded by hispidulin (Figure 3c).

### 2.5. Hispidulin Downregulates the P. gingivalis LPS-Induced Transcriptional Activation of ICAM-1 through NF-κB Activation

NF-κB mediates the induction of pro-inflammatory cytokines and cell adhesion molecules, thereby leading to pro-inflammatory responses in vascular endothelial cells [23]. Notably, the ICAM-1 promoter has binding sites for NF-κB. We performed luciferase promoter assays to determine whether hispidulin regulated ICAM-1 transcriptional activity. As depicted in Figure 4a, the full-length promoter regions of ICAM-1 (1.3 kb) in the luciferase reporter construct contained the NF-κB, ARE, and TATA binding sites [24]. The ICAM-1 truncated promoter construct contained a proximal NF-κB binding motif (Figure 4a). These constructs were introduced into HUVECs, which were then treated with *P. gingivalis* LPS, either with or without hispidulin, whereas *P. gingivalis* LPS enhanced the reporter activity of the truncated ICAM-1 promoter; this increase was attenuated via hispidulin treatment (Figure 4a). These results were similar to those obtained with the full-length ICAM-1 promoter, indicating that the NF-κB binding site played an essential role in mediating the effect of hispidulin on *P. gingivalis* LPS-activated promoters. Inflammatory stimuli promote the translocation of the p65 subunit of NF-κB into the nucleus and its subsequent binding to cognate DNA binding sites to upregulate several inflammation-related genes [25]. Immunocytochemical analysis demonstrated the localization of p65 (35.7%) within the nucleus of HUVECs after *P. gingivalis* LPS stimulation. In contrast, exposure to *P. gingivalis* LPS and hispidulin decreased nuclear p65 (11.4%) accumulation (Figure 4b).

### 2.6. Hispidulin Downregulates P. gingivalis LPS-Induced MAPKs and AKT in Vascular Endothelial Cells

*P. gingivalis* LPS triggers MAPK signaling, including extracellular signal-regulated kinase (ERK), c-Jun N-terminal kinase (JNK), and p38MAPK in human gingival fibroblasts, leading to increased cytokine production [26]. The AKT signaling pathway plays an important role in regulating the survival of human gingival epithelial cells [27]. We examined the impact of hispidulin on the MAPK and AKT pathways in *P. gingivalis* LPS-stimulated endothelial cells. HUVECs were exposed to *P. gingivalis* LPS either alone or in combination with hispidulin. Subsequently, the phosphorylation of ERK, JNK, p38MAPK, and Akt was assessed using Western blot analysis. The addition of hispidulin reduced the *P. gingivalis* LPS-stimulated phosphorylation of ERK, JNK, and p38MAPK, and significantly inhibited that of AKT in HUVECs (Figure 4c). 

## 3. Discussion

Hispidulin is a natural flavonoid with appealing anti-inflammatory, antioxidant, neuroprotective, anticancer, anti-diabetic, and anti-microbial activities and exerts beneficial effects on various inflammatory diseases, including allergic inflammation, atopic dermatitis, and neuroinflammation [28,29,30]. Hispidulin has been reported to downregulate LPS-induced inflammatory responses in vitro and in vivo. Hispidulin inhibits the LPS-induced production of tumor necrosis factor-alpha, IL-1β, and IL-6 in microglial cells by suppressing the activation of the NF-κB, AKT, and STAT3 signaling pathways [16]. Additionally, hispidulin attenuates LPS-induced acute kidney injury in mice by reducing the expression of pro-inflammatory cytokines and Toll-like receptor 4 (TLR4) [31]. Targeting the signaling pathway mediated by TLR4 is considered a strategy for improving diseases associated with inflammation [32]. Hispidulin has been reported to reduce cytokine production in endotoxin-induced kidney injury, which is dependent on TLR4, via modulating NF-κB and MAPK signaling pathways [16]. TLR4 signaling plays a crucial role in the recognition of *P. gingivalis* LPS, leading to the activation of downstream signaling pathways that ultimately produce pro-inflammatory cytokines and chemokines [33]. The stimulation of TLR4 by *P. gingivalis* LPS induces the production of proinflammatory cytokines in endothelial cells [34]. Additionally, we observed that hispidulin downregulates the *P. gingivalis* LPS-induced mRNA expression of TLR4 in endothelial cells (Appendix A). In this study, we demonstrate that hispidulin inhibits the NF-қB, AKT, and MAPK (ERK, JNK, and p38MAPK) signaling pathways activated by *P. gingivalis* LPS, along with the downstream production of pro-inflammatory cytokines and cell adhesion molecules in vascular endothelial cells. Thus, our results suggest that the negative effect of hispidulin on the inflammatory response triggered by *P. gingivalis* LPS may be due to the inhibition of the TLR4-dependent signaling pathway. 

Recently, natural phenolic compounds capable of modulating host inflammatory responses have received attention as effective tools for managing periodontal disease [7]. Myricetin and apigenin inhibit the expression of inflammatory cytokines and matrix metalloproteinases in different types of cells in the periodontium, such as gingival fibroblasts and periodontal ligament cells [35,36]. Curcumin and quercetin alleviated experimental periodontitis in animal models by reducing gingival inflammation and attenuating alveolar bone loss [37,38]. Alveolar bone loss is a characteristic of the progression of periodontitis, making its suppression a crucial clinical objective in the treatment of periodontal disease [39]. Aberrant angiogenesis is known to contribute to the pathogenesis of periodontitis in periapical lesions [40]. Hispidulin blocks RANKL-induced osteoclastogenesis in vitro and LPS-induced bone resorption in vivo [41]. Furthermore, hispidulin inhibits angiogenesis by suppressing the migration and tubular formation of endothelial cells [17]. Our preliminary results suggest that hispidulin lowers the expression of pro-inflammatory cytokines in human gingival fibroblasts. Based on these findings, it is expected that hispidulin may have the potential to prevent and inhibit the progression of periodontal disease. This possibility is currently under investigation.

Atherosclerosis is a chronic inflammatory disease of the arterial wall, leading to plaque formation within the arteries [42]. Vascular endothelial cells, which line the inner surface of blood vessels, are essential for the development and progression of atherosclerosis [43]. Once the endothelial barrier is compromised, inflammatory cells enter the arterial wall and contribute to the formation of atherosclerotic plaques [44]. In this study, we demonstrate that hispidulin reduces monocyte adhesion to *P. gingivalis* LPS-activated endothelial cells and their transmigration across the endothelial layer. The presence of *P. gingivalis* or *P. gingivalis* LPS may be an additional risk factor that exacerbates the progression of atherosclerosis [45]. *P. gingivalis* LPS-accelerated atherosclerosis typically progresses through several stages, including foam cell formation, vascular smooth muscle cell proliferation/migration, plaque rupture, and endothelial dysfunction [46]. Further investigations are ongoing to determine whether hispidulin inhibits several atherosclerotic properties in an established *P. gingivalis* LPS-accelerated atherosclerosis model in ApoE−/− mice.

In conclusion, our results show that hispidulin suppresses *P. gingivalis* LPS-induced adhesion and the transmigration of monocytes through the vascular endothelium and the expression of pro-inflammatory cytokines and cell adhesion molecules. This is achieved by inhibiting NF-κB and blocking MAPK and AKT activations. Our findings indicate that hispidulin has beneficial effects in managing and treating periodontal pathogen-associated atherosclerosis. 

## 4. Materials and Methods

### 4.1. Reagents 

Hispidulin (#SML0582) and MTT (#M5655) were acquired from Sigma-Aldrich (St. Louis, MO, USA). Antibodies against human ICAM-1 and α-tubulin were obtained from Santa Cruz Biotechnology (#SC-8439, Dallas, TX, USA) and Bioworld Technology (#BS1699, St. Louis Park, MN, USA), respectively. Antibodies against human ERK (#9102), phospho-ERK (#9106), JNK (#9252), phospho-JNK (#4668), p38MAPK (#9212), phospho-p38MAPK (#9215), AKT (#9272), and phospho-AKT (#9271) were acquired from Cell Signaling Technology (Danvers, MA, USA).

### 4.2. Cell Culture 

HUVECs were purchased from CLONETICS (Basel, Switzerland), plated on 0.2% gelatin-coated dishes, and grown in endothelial cell basal medium-2 (EBM-2; Lonza, Basel, Switzerland) supplemented with EGM-2 SingleQuots™ (Lonza) at 37 °C in humidified air with 5% CO_2_. HUVECs from passage 2 were seeded at passages 4–7 for subsequent experiments. Human THP-1 monocytes were obtained from the Korea Cell Line Bank and cultured at the Roswell Park Memorial Institute (RPMI) 1640 (Gibco, Billings, MT, USA) with 10% fetal bovine serum (Gibco), 1% penicillin-streptomycin (Gibco), and 5 μg/mL Plasmocin^®^ (Invitrogen, Carlsbad, CA, USA).

### 4.3. Cell Proliferation Assay

HUVECs (2 × 10^4^ cells per well) were seeded on 48-well plates and incubated for 24, 48, or 72 h at 37 °C. At the end of the culture period, the cells were incubated in 500 μL new medium containing 0.5 mg/mL MTT (Sigma-Aldrich) for 4 h. The medium was changed with 200 μL dimethyl sulfoxide (Sigma-Aldrich) per well for 3 min. The resulting blue formazan product was measured at 540 nm with a multiwell plate reader (Allsheng, Hangzhou, China).

### 4.4. Reverse Transcription Quantitative Polymerase Chain Reaction (RT-qPCR)

Total RNA was extracted from HUVECs using a RiboEx kit (#301-001, GeneAll, Seoul, Republic of Korea) and reverse-transcribed with a reverse transcription kit (#A3500, Promega, Madison, WI, USA) according to the manufacturer’s instruction, followed by an RT-qPCR with SYBR Green premix (#RT-500, Enzynomix, Daejeon, Republic of Korea). The following oligonucleotide primers were used: β-actin 5′-ACTCTTCCAGCCTTCCTTCC-3′ and 5′-TGTTGGCGTACAGGTCTTTG-3′; monocyte chemoattractant protein-1 (MCP-1) 5′-ACTCTCGCCTCCAGCATGAA-3′ and 5′-TTGATTGCATCTGGCTGAGC-3′; Interleukin (IL)-1β 5′-GACCTGGACCTCTGCCCTCT-3′ and 5′-CTGCCTGAAGCCCTTGCTGT-3′; IL-8 5′-CTGGCCGTGGCTCTCTTG-3′ and 5′-CCTTGGCAAAACTGCACCTT-3′; ICAM-1 5′-CCCCACCATGAGGACATACA-3′ and 5′-GTGTGGGCCTTTGTGTTTTG-3′. Cycling parameters included one cycle at 95 °C for 10 min, followed by amplification for 40 cycles at 95 °C for 15 s, 60 °C for 60 s, and 72 °C for 7 s. The RT-qPCR analyses were performed using a 7500 Real-Time PCR system (Applied Biosystems, Foster City, CA, USA). Relative gene expression was calculated using the comparative C(T) method also referred to as the 2 (-DeltaDeltaC(T)) method [47]. The relative fold change of the target genes was normalized to the level of β-actin and the control. 

### 4.5. Western Immunoblot Analysis

An amount of 30 μg of total protein per lane were loaded on 10% bis-acrylamide gels, and electrophoresis was performed at 80 volts for 2 h. After being transferred to a nitrocellulose membrane (Amersham Pharmacia Biotech, Uppsala, Sweden), the membranes were blocked in 5% skim milk for 1 h under agitation and probed with human α -tubulin (diluted 1:10,000), ICAM-1 (diluted 1:1000), ERK (diluted 1:5000), phospho-ERK (diluted 1:2000), JNK (diluted 1:5000), phospho-JNK (diluted 1:2000), p38MAPK (diluted 1:5000), phospho-p38MAPK (diluted 1:2000), AKT (diluted 1:5000), and phospho-AKT (diluted 1:2000) antibodies for 16 h at 4 °C. After incubation with primary antibodies, the membranes were washed three times for 10 min with phosphate-buffered saline (PBS) and incubated for 1 h at room temperature with horseradish peroxidase-conjugated goat-anti-rabbit IgG (diluted 1:10,000, #ADI-SAB-500-J, ENZO Life Science, Farmingdale, MY, USA) or goat-anti-mouse IgG (diluted 1:10,000, #ADI-SAB-100-J, ENZO Life Science) as the secondary antibodies. The signal was developed using an enhanced chemiluminescence solution (Amersham Pharmacia Biotech) and visualized on an Azure 300 Imaging system (Azure Biosystems, Dublin, CA, USA).

### 4.6. Flow Cytometry Analysis

HUVECs (1 × 10^6^ cells per well) were seeded in a 60 mm dish and exposed to *P. gingivalis* LPS (5 µg/mL) alone, hispidulin alone (5 µM), or a combination of *P. gingivalis* LPS and hispidulin for 16 h. HUVECs were washed twice with PBS and incubated with phycoerythrin-conjugated anti-human CD54 (ICAM-1, 50 μg/mL, #555511, BD Biosciences, Bedford, MA, USA) at 4 °C. After 1 h, the cells were rinsed twice with 5 mL PBS and suspended in 500 µL PBS. The cells transferred into the FACS tube were analyzed using flow cytometry (BD Biosciences, Franklin Lakes, NJ, USA). 

### 4.7. In Vitro Monocyte Adhesion Assay

HUVECs (5 × 10^4^) were seeded on 24-well plates and exposed to *P. gingivalis* LPS (5 µg/mL) alone, hispidulin alone (5 µM), or a combination of *P. gingivalis* LPS and hispidulin for 16 h. Confluent monolayers of HUVECs were incubated with THP-1 (1 × 10^5^ cells per well) cells for 1 h. Before their addition, THP-1 cells were stained with 5 μg/mL calcein-AM (#C1430, Invitrogen) for 30 min. Non-adherent monocytes were carefully washed with PBS. Images of the adherent cells were captured in three separate random fields in each well using a fluorescence microscope (Korea Lab Tech, Seungnam, Republic of Korea). 

### 4.8. Ex Vivo Monocyte Adhesion Assay

Six-week-old male Sprague Dawley rats were acquired from Koatech (Pyeongtaek, Republic of Korea). The aortas were exposed to *P. gingivalis* LPS (5 µg/mL) alone, hispidulin alone (5 µM), or a combination of *P. gingivalis* LPS and hispidulin for 16 h. After incubation, calcein-AM (5 μg/mL)-labeled THP-1 (5 × 10^5^ cells per well) cells were added to the aortas for 1 h. After incubation, unbound monocytes were gently washed three times with PBS. In contrast, adherent cells were counted in three random fields using a fluorescence microscope. 

### 4.9. Transmigration Assay

HUVECs (5 × 10^4^) were added in the upper chamber of transwells with 8 µm pore-size membrane inserts (Costar, Corning, NY, USA) and cultured for 24 h to form a confluent monolayer. After incubation, calcein-AM (5 μg/mL)-labeled THP-1 cells were added to the upper chamber and allowed to migrate through the HUVEC monolayer to the lower chamber for 24 h. These cells were exposed to *P. gingivalis* LPS (5 µg/mL) alone, hispidulin alone (5 µM), or a combination of *P. gingivalis* LPS and hispidulin in EBM-2 medium supplemented with EGM-2 SingleQuots™. The lower chamber was filled with 500 μL RPMI 1640 medium. Images were captured in three random fields using a fluorescence microscope. 

### 4.10. Transient Transfection and Reporter Gene Analysis

HUVECs (1 × 10^6^) were transfected with 3 μg of the plasmid DNA using Amaxa Nucleofector II (Lonza). After transfection, HUVECs were exposed to *P. gingivalis* LPS (5 µg/mL) alone, hispidulin alone (5 µM), or a combination of *P. gingivalis* LPS and hispidulin for 16 h. After washing twice with PBS, the cells were resuspended in reporter lysis buffer (Promega) and incubated for 10 min. To separate the cell debris, the lysate was subjected to centrifugation at 12,000× *g* for 5 min. The supernatants transferred to a new tube were added to the luciferase assay substrate (Promega) and the luminescence of the samples was measured on a 20/20 Luminometer (Turner Biosystems, Sunnyvale, CA, USA). To standardize transfection efficiency and protein input, β-Galactosidase activity served as a normalization factor. The cell extract was mixed with O-nitrophenyl-β-D-Galactopyranoside (ONPG) solution. After 30 min of incubation, the absorbance of the mixture was determined at a wavelength of 420 nm. The ICAM-1 luciferase reporter constructs with the full-length (−1350 to +45 bp) and truncated forms (−485 to +45 bp), provided by Dr. Young-Guen Kwon (Yonsei University, Seoul, Republic of Korea), were used as previously described [24].

### 4.11. Immunocytochemistry

After 30 min pretreatment with hispidulin (5 µM), HUVECs were exposed to *P. gingivalis* LPS (5 µg/mL) for 1 h and then fixed in 4% paraformaldehyde for 10 min. Subsequently, the cells were blocked using a 0.5% Triton X-100/PBS and 5% normal goat serum (#S-1000-20, Vector Labs, Burlingame, CA, USA) and reacted with the primary antibody against NF-κB p65 (diluted 1:100, #SC-8008, Santa Cruz Biotechnology) for 1 h. After incubation with the primary antibody, the cells were rinsed three times for 10 min with PBS and incubated with the Alexa^®^ 488-conjugated secondary antibody (diluted 1:200, #A11001, Invitrogen) for 1 h in the dark. Coverslips were mounted with DAPI-containing Vectastain (#H-1200, Vector Laboratories). Cell analysis was conducted using a confocal microscope (LSM900; Zeiss, Oberkochen, Germany).

### 4.12. Statistical Analysis

Data represent the mean and standard deviation of at least three independent experiments. Data were subjected to one-way analysis of variance with Tukey’s honest significant difference post hoc test and Student’s *t*-test using IBM SPSS v27 (Chicago, IL, USA).

## Figures and Tables

**Figure 1 molecules-28-06717-f001:**
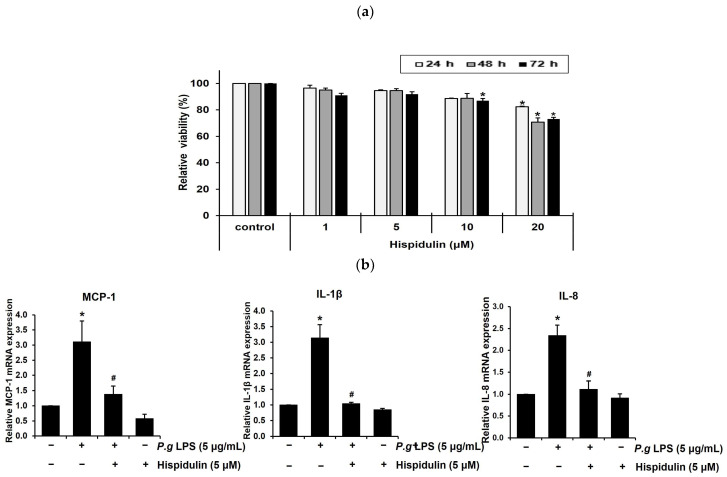
The effect of hispidulin on the expression of pro-inflammatory cytokines in HUVECs. (**a**) HUVECs were treated with various concentrations of hispidulin for the indicated times. After treatment, cell proliferation was conducted using the MTT assay. * *p* < 0.01 compared to that in control. (**b**) HUVECs were exposed to *P. gingivalis* LPS (5 µg/mL) alone, hispidulin alone (5 µM), or a combination of *P. gingivalis* LPS and hispidulin for 16 h. The expression of MCP-1, IL-1β, and IL-8 was analyzed using RT-qPCR. The control level was established at 1.0, and the measurements were adjusted relative to β-actin for normalization. * *p* < 0.01 compared to that of control. ^#^
*p* < 0.01 compared to that of *P. gingivalis* LPS. Data shown are the mean ± SD, obtained from at least three independent experiments.

**Figure 2 molecules-28-06717-f002:**
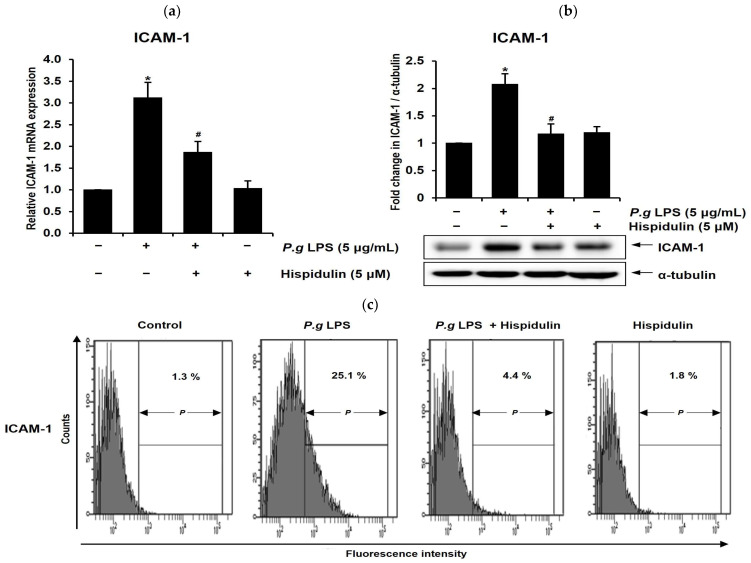
Hispidulin reduces ICAM-1 expression in *P. gingivalis* LPS-induced HUVECs. (**a**,**b**) HUVECs were exposed to *P. gingivalis* LPS (5 µg/mL) alone, hispidulin alone (5 µM), or a combination of *P. gingivalis* LPS and hispidulin for 16 h, and the expression of ICAM-1 was analyzed using RT-qPCR and Western blotting. The control level was established at 1.0, and the measurements were adjusted relative to β-actin for normalization (**a**) or α-tubulin as the internal control. The quantification of the ICAM-1 protein level was obtained with densitometry and normalized to α-tubulin (**b**). * *p* < 0.01 compared to that of control. ^#^
*p* < 0.01 compared to that of *P. gingivalis* LPS. Data shown are the mean ± SD, obtained from at least three independent experiments. (**c**) HUVECs were exposed to *P. gingivalis* LPS (5 µg/mL) alone, hispidulin alone (5 µM), or a combination of *P. gingivalis* LPS and hispidulin for 16 h. ICAM-1 (CD54) was quantified using flow cytometry.

**Figure 3 molecules-28-06717-f003:**
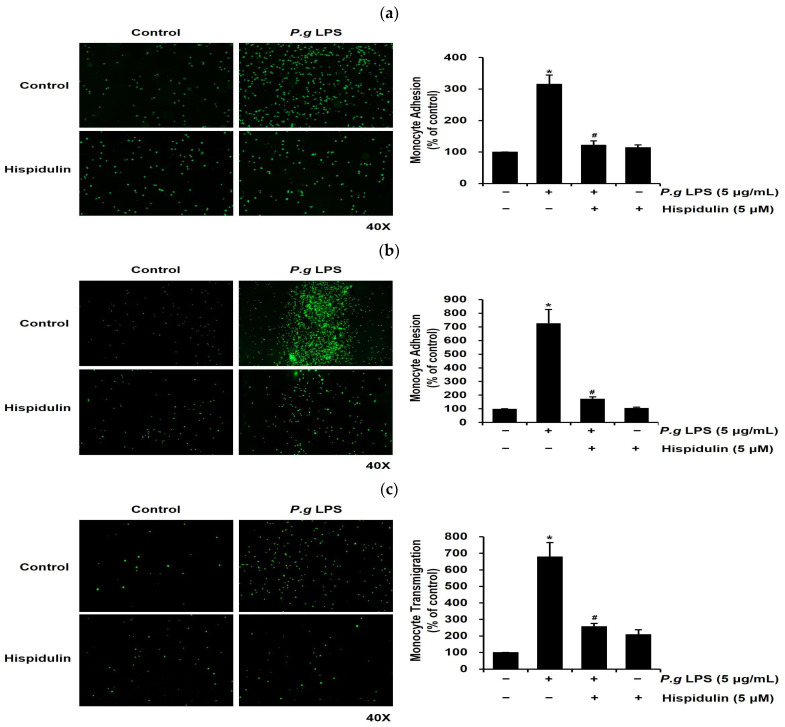
Hispidulin decreases *P. gingivalis* LPS-induced monocyte adhesion to the endothelium in vitro and ex vivo and the transendothelial migration of monocytes. (**a**) HUVECs were exposed to *P. gingivalis* LPS (5 µg/mL) alone, hispidulin alone (5 µM), or a combination of *P. gingivalis* LPS and hispidulin for 16 h. Calcein-AM-labeled THP-1 cells (green) were identified using fluorescence microscopy at 40× magnification. Adherent monocytes were counted in three fields for each set of four wells and normalized to the control group. * *p* < 0.01 compared to that of the control. ^#^
*p* < 0.01 compared to that of *P. gingivalis* LPS. (**b**) The aorta isolated from Sprague Dawley rats was exposed to *P. gingivalis* LPS (5 µg/mL) alone, hispidulin alone (5 µM), or a combination of *P. gingivalis* LPS and hispidulin for 16 h. At the end of incubation, THP-1 cells were stained with calcein-AM for 30 min and then observed using fluorescence microscopy at 40× magnification. * *p* < 0.01 compared to that of the control. ^#^
*p* < 0.01 compared to that of *P. gingivalis* LPS. (**c**) To examine whether THP-1 cells migrated across endothelial cells and reached the lower chamber, HUVECs were exposed to *P. gingivalis* LPS (5 µg/mL) alone, hispidulin alone (5 µM), or a combination of *P. gingivalis* LPS and hispidulin for 16 h and co-cultured with calcein-AM-labeled THP-1 monocytes in the upper chamber of transwells. The number of THP-1 cells (green) in the lower chamber of the transwell was counted from randomly acquired images (40× magnification). * *p* < 0.01 compared to that of the control. ^#^
*p* < 0.01 compared to that of *P. gingivalis* LPS. Data shown are the mean ± SD, obtained from at least three independent experiments.

**Figure 4 molecules-28-06717-f004:**
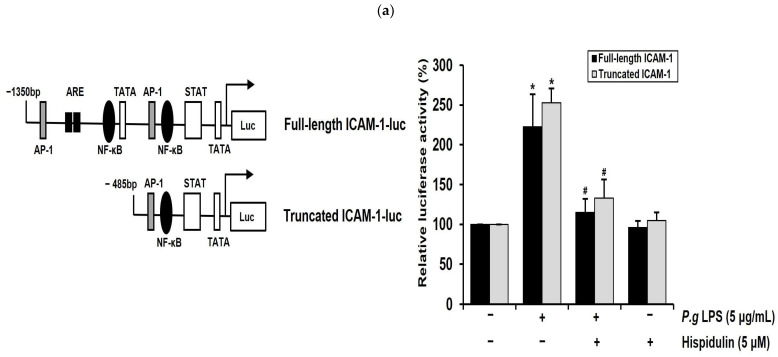
Hispidulin inhibits *P. gingivalis* LPS-induced NF-κB-dependent ICAM-1 promoter activity, MAPKs, and AKT signaling. (**a**) HUVECs were transiently transfected with the full-length and truncated promoters of the ICAM-1 gene. Transfected HUVECs were exposed to *P. gingivalis* LPS (5 µg/mL) alone, hispidulin (5 µM) alone, or a combination of *P. gingivalis* LPS and hispidulin for 16 h. * *p* < 0.01 compared to that of the control. ^#^
*p* < 0.01 compared to that of *P. gingivalis* LPS. Data shown are the mean ± SD, obtained from at least three independent experiments. (**b**) HUVECs were pretreated with hispidulin (5 µM) for 30 min and then incubated for 1 h with *P. gingivalis* LPS (5 µg/mL). NF-κB p65 (green) localization in the nuclei (blue) was observed. (**c**) HUVECs were pretreated for 30 min with or without hispidulin (5 µM) before stimulation with *P. gingivalis* LPS (5 μg/mL) for 10 min. Anti-phospho-ERK, anti-ERK, anti-phospho-JNK, anti-JNK, anti-phospho-p38MAPK, anti-p38MAPK, anti-phospho-AKT, and anti-AKT antibodies were used to probe the Western blots. α-tubulin served as the internal control.

## Data Availability

Data is available as request to corresponding author (M.-K.B.) upon reasonable request.

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
