# Peer review of "Hispidulin Inhibits the Vascular Inflammation Triggered by Porphyromonas gingivalis Lipopolysaccharide"

_molecules, 2023, doi:10.3390/molecules28186717_

Round 1

Reviewer 1 Report

Dear

Akkasubha Kotchabhakdi

Assistant Editor

Thank you very much for allowing me to review the article with title “Hispidulin inhibits vascular inflammation triggered by Porphyromonas gingivalis lipopolysaccharide” by the authors Yeon Kim et al. Where the aim of this study was to explore whether hispidulin could inhibit endothelial inflammation triggered by Porphyromonas gingivalis (P. gingivalis) lipopolysaccharide (LPS) and the authors concluded that hispidulin suppresses vascular inflammation induced by P. gingivalis LPS. Through of the preventing the adhesion of monocytes to the vascular endothelium and migration and inhibits NF-Ò›B, MAPKs, and AKT signaling in endothelial cells.

The article is well structured from the summary, the introduction is solid and supports the experimental work, in turn the findings in the results adequately support the discussion and the conclusion, the article reads and understands it satisfactorily. The methods are more adequate to support the experiments and results obtained.

he original western blots are the representative ones shown in the different figures

I only have some suggestions that could help improve this article.

In figure 1 the authors could show each of the figures larger, because it is difficult to see the units and the asterisks of the significances. or they could substitute the values obtained in these graphs and generate a table (figure 1B) and in this way all the variables of the three histograms would be presented in a table.

In figure 2 the authors could show in a single figure both the relative density of ICAM-1 and in the lower part of the histogram the representative western blots, in this way the comparison for the reader would be easier.

They could add the percentage of the polyacrylamide gel

Could you add the software of the statistical program that you used to itself as the p? which is taken as significant

Best Regards

Reviewer

Author Response

We would like to express our gratitude to the reviewer for valuable feedback, which has significantly enhanced the quality of our manuscript. We have revised our manuscript based on the comments as described below.

RESPONSE TO REVIEWER 1

REVIEWER COMMENTS:

Thank you very much for allowing me to review the article with title “Hispidulin inhibits vascular inflammation triggered by Porphyromonas gingivalis lipopolysaccharide” by the authors Yeon Kim et al. Where the aim of this study was to explore whether hispidulin could inhibit endothelial inflammation triggered by Porphyromonas gingivalis (P. gingivalis) lipopolysaccharide (LPS) and the authors concluded that hispidulin suppresses vascular inflammation induced by P. gingivalis LPS. Through of the preventing the adhesion of monocytes to the vascular endothelium and migration and inhibits NF-Ò›B, MAPKs, and AKT signaling in endothelial cells. The article is well structured from the summary, the introduction is solid and supports the experimental work, in turn the findings in the results adequately support the discussion and the conclusion, the article reads and understands it satisfactorily. The methods are more adequate to support the experiments and results obtained. The original western blots are the representative ones shown in the different figures.

I only have some suggestions that could help improve this article.

Point 1. In figure 1 the authors could show each of the figures larger, because it is difficult to see the units and the asterisks of the significances. or they could substitute the values obtained in these graphs and generate a table (figure 1B) and in this way all the variables of the three histograms would be presented in a table.

Response 1: Based on the reviewer’s recommendations, we have made modified to Figure 1 to improve its clarity for readers. We have enlarged the figure and adjusted the units and significance asterisks to enhance its readability in the updated Figure 1 of the revised manuscript. Please refer to Figure 1 in the revised manuscript.

Point 2-1. In figure 2 the authors could show in a single figure both the relative density of ICAM-1 and in the lower part of the histogram the representative western blots, in this way the comparison for the reader would be easier.

Response 2-1: What Figure 2a illustrates is not the relative density of ICAM-1 protein but rather the expression level of ICAM-1 mRNA using real-time qPCR. The relative density of ICAM-1 protein band from western blots were quantified, and the resulting quantified plot, based on independent experiments, was newly incorporated into the upper part of Figure 2b of the revised manuscript.

Point 2-2. They could add the percentage of the polyacrylamide gel.

Response 2-2: In this study, we used 10% polyacrylamide gels for western blot analysis. Following your suggestion, we have included the polyacrylamide gel percentage in the “Materials and Methods” section of the revised manuscript.

Point 3. Could you add the software of the statistical program that you used to itself as the p? which is taken as significant.

Response 3: We performed Tukey's Honestly Significant Difference (HSD) test to compare the means among the four groups. The one-way analysis of variance (ANOVA) with a post hoc Tukey HSD test revealed statistical differences between the data groups, and we reported the respective p-values based on these results. Furthermore, we have added more detailed information about the statistical software in the “Materials and Methods” section of the revised manuscript.

.

Author Response

We would like to express our gratitude to the reviewer for valuable feedback, which has significantly enhanced the quality of our manuscript. We have revised our manuscript based on the comments as described below.

RESPONSE TO REVIEWER 2

REVIEWER COMMENTS:

This study aims to investigate the effects of hispidulin on HUVEC population treated by P. gingivalis LPS. The anti-bacterial, anti-fungal and anti-inflammatory activities of the polyphenolic compound hispidulin was already presented in considerable amounts of in vivo/in vitro studies. Similarly, the contribution of Porphyromonas gingivalis to chronic inflammatory disease has been already widely published. Reading the manuscript many concerns emerge about the experimental design and the presentation / interpretation of the data sets.

Response:

Our research presents the first evidence of hispidulin's suppressive effect on vascular inflammation triggered by the lipopolysaccharide of Porphyromonas gingivalis. In response to your comment, we've enriched the 'Introduction' with more comprehensive background information pertinent to our study. We've also honed the 'Discussion' section, elucidating the anti-inflammatory properties of hispidulin and its potential to impede the progression of periodontal disease. Additionally, we've augmented the 'Results' and ‘Figure legend’ sections with detailed outcomes and enhanced the 'Materials & Methods' section to offer a more thorough account of the experimental procedures." Please find below a point-by-point response to all comments.

Specific comments:

Point 1. The introduction part of the manuscript is incomplete, not only in its subject matter but also in the citation of the topics mentioned. There is a lack of some basic information about the topic, though some of them are presented later on in the result section. They should be referred in the introduction or discussion part.

Response 1: In response to your comment, we have revised the 'Introduction' section to include descriptions that provide a comprehensive background for the study (line 48-51; line 60-62; line 64-66) and have incorporated additional references to support our additions."

Point 2. Result section is full of discrepancies. I would like to highlight only a few of them;

Point 2-1. The figures and their captions (in many cases even the protocols in the Materials and Methods section) are contradictory. In the figure legends authors refer for the experimental designe as "HUVECs were treated with P. gingivalis LPS (5 μg/mL) alone or in combination with hispidulin (5 μM) for 16 h". However, there are also data presented on hispidulin treatment alone (Figures 1b; 2 a, b, c; 3 a,b,c… ). Figure legends should be changed accordingly.

Response 2-1: As the reviewer pointed out, we have thoroughly reviewed all the figures and their captions in our manuscript for any contradictions. Additionally, we corrected the existing sentence to provide clearer identification of missing groups in the "Figure legends" and “Materials and Methods” section of the revised manuscript.

Old: HUVECs were treated with P. gingivalis LPS (5 µg/mL) alone or in combination with hispidulin (5 µM) for 16 h.

New: HUVECs were exposed to P. gingivalis LPS (5 µg/mL) alone, hispidulin alone (5 µM), or a combination of P. gingivalis LPS and hispidulin for 16 h.

Point 2-2. In the Materials and Methods section authors list 3 different concentration for P. gingivalis LPS treatment 5mg/ml (M&M: 4.7), 5g/ml (M&M: 4.9) and 10 mg/ml (M&M: 4.6). What justified the use of different concentrations?

Response 2-2: We carefully reexamined the manuscript based on the reviewers' comments. As a result, we identified a serious typographical error in the notation of the P. gingivalis LPS concentration in the Materials and Methods section. The correct concentration of P. gingivalis LPS used in this study was 5 µg/ml, and all incorrectly stated concentration values (5 g/ml or 10 µg/ml) have been rectified to 5 µg/ml in the revised manuscript. We would like to thank the reviewers for helping us improve the accuracy of our research by identifying potential serious errors in the paper in advance.

Point 2-3. What was the reason for the change the order of hispidulin and P. gingivalis LPS treatment? Similarly, what was the reason for the alteration of the timing in the case NF-κB p65 experiments; “HUVECs were pretreated with hispidulin (5 μM) for 30 min and then incubated for 1 h with P. gingivalis LPS (5 μg/mL). NF-κB p65 (green) localization in the nuclei (blue) was observed. (c) HUVECs were pretreated for 30 min with or without hispidulin (5 μM) before stimulation with P. gingivalis LPS (5 μg/mL) for 10 min.”

Response 2-3: Based on previous studies [Park, H.J, et al., “Resveratrol inhibits Porphyromonas gingivalis lipopolysaccharide-induced endothelial adhesion molecule expression by suppressing NF-kappaB activation.” Arch. Pharm. Res. 32: 583-591, 2009], to determine the optimal conditions for the translocation of NF-κB p65, HUVECs were exposed to P. gingivalis LPS for various times (up to 2 hours in 30 minutes increments), and hispidulin was also tested using both co-treatment and pre-treatment methods. As a result, when HUVECs were pretreated with hispidulin and then exposed to P. gingivalis LPS for 1 hr, the inhibitory effect of hispidulin on the P. gingivalis LPS-induced NF-κB p65 translocation was most clearly confirmed compared to the other conditions. Therefore, we have selected these results as the final data for the paper. The Figure 4c mentioned by the reviewer represents the results confirming the phosphorylation levels of the ERK, AKT, p38MAPK, and JNK signaling pathways. Since these signaling pathways differ from the NF-κB p65 translocation pathway, there might be variations in the timing of activation for each signaling pathway. This is in line with findings from other studies [Ying Wang et al., Atorvastatin suppresses LPS-induced rapid upregulation of Toll-like receptor 4 and its signaling pathway in endothelial cells. Am J Physiol Heart Circ Physiol 300: H1743-1752, 2011; Qiang Chen, et al., Mycoepoxydiene inhibits lipopolysaccharide-induced inflammatory responses through the suppression of TRAF6 polyubiquitination. PLoS One 7(9): e44890. 2012].

Point 2-4. (Figure 2) To be honest, the purpose of using flow cytometry is not entirely clear for me, since the dynamic of hispidulin effect on the P. gingivalis LPS induced cell population was already followed by qPCR and Western blot experiments. What was the reason to apply P. gingivalis LPS in different concentration (10 μg/mL) for flow cytometry?

Response 2-4: Analysis of cell surface proteins by flow cytometry is common due to ICAM-1 being a cell surface glycoprotein expressed at low basal levels in endothelial cells but upregulated in response to inflammatory stimuli [Bei Y.R, et al., “ EPSTI1 promotes monocyte adhesion to endothelial cells in vitro via upregulating VCAM-1 and ICAM-1 expression. Acta Pharmacol Sin 44(1):71-80. 2023]. Flow cytometry also allows for multiplexed and quantitative analysis of signaling events, providing higher sensitivity and precision compared to Western blotting. Therefore, despite having results from mRNA and Western blot analysis, we utilized flow cytometry to ensure the accuracy of our findings. The description of the P. gingivalis LPS concentration used in flow cytometry has been previously provided in Response 2-2.

Point 2-5. Figure 4 (b) what was the frequency (in %) of nuclear localisation of NF-κB p65?

Response 2-5: As a reviewer suggested, we conducted a quantitative analysis of the translocation of NF-κB p65 and have included it as Table 1 for review purposes only (Please see an attached file). We found that the translocation frequency was as follows: 6.13 % (± 0.21) in the control group, 35.69 % (± 7.17) in the P.gingivalis LPS alone group, 11.40 % (± 2.15) in the P.gingivalis LPS+ Hispidulin group, and 6.65 % (± 1.88) in the Hispidulin alone group. Additionally, we have briefly mentioned the explanation for the quantitative analysis of NF-κB p65 translocation in the “Results” section of our revised manuscript.

Point 2-6. In the authors interpretation hispidulin treatment decreased the expression level of the investigated mRNAs and proteins induced by P. gingivalis LPS. Based on the protocol, cells were treated with P. gingivalis LPS alone or in combination with hispidulin. Isn’t it possible that hispidulin inhibits the induction of mRNA/protein synthesis by P. gingivalis LPS?

Response 2-6: To investigate whether hispidulin suppressed the mRNA and protein expression of inflammatory markers induced by P. gingivalis in HUVECs, we organized groups consisting of control, P. gingivalis LPS alone, P. gingivalis LPS + Hispidulin, and Hispidulin alone, and subsequently conducted the experiment. As a result, we demonstrated that the mRNA and protein expressions of inflammatory markers, which had increased due to treatment with P. gingivalis LPS alone, were completely reduced when P. gingivalis LPS and hispidulin were co-administered. When hispidulin was administered alone, it had no effect on the expression of mRNA and protein. These results fully corroborate the hypothesis that hispidulin inhibits the mRNA and protein expression of inflammatory markers induced by P. gingivalis LPS.

In addition, we showed in this study that hispidulin downregulates P. gingivalis LPS-induced transcriptional activation of ICAM-1 through NF-κB activation. Regulation of ICAM-1 is cell-type specific and predominantly occurs at the transcriptional level; however, post-transcriptional regulation is also known to be involved in ICAM-1 gene expression [Singh M, et al., Gene regulation of intracellular adhesion molecule-1 (ICAM-1): A molecule with multiple functions, Immunol Lett. 240:123-136, 2021]. Therefore, we plan to further investigate whether P. gingivalis LPS regulates ICAM-1 expression in vascular endothelial cells at the post-transcriptional level, particularly elucidating the impact of hispidulin on post-transcriptional regulation of ICAM-1 expression.

Point 3. Discussion section is the most critical part of the manuscript. First of all, it is a modest discussion of only a partial set of the reviewed data. Secondly authors drown detailed conclusion from unpublished data set, introduced only in this section. “Additionally, we observed that hispidulin downregulates P. gingivalis LPS-induced mRNA expression of TLR4 in endothelial cells (data not shown)” “Our preliminary results suggest that hispidulin lowers the expression of pro-inflammatory cytokines in human gingival fibroblasts (data not shown), hinting at its anti-inflammatory role in periodontitis.” Numerous other things should /could be mentioned in this section.

Response 3: In response to the reviewer's feedback, we have made revisions to the "Discussion" section to improve the clarity of the explanation regarding the anti-inflammatory action of hispidulin (lines 222-226). Furthermore, in accordance with this update, we have included a new figure illustrating the mRNA expression of TLR4 in the "Supplementary Data" section of the revised manuscript. (In the original manuscript, this was referred to as unpublished data). In addition, we have incorporated new sentences into the 'Discussion' section (lines 244-249) that provide information about the potential role of hispidulin in inhibiting the progression of periodontal disease. And we have included additional references to support our statements."

Point 4. Materials and Methods

Beyond the above mentioned criticism this part of the MS raises many questions due to many inaccuracies and lack of proper information. Again, I will highlight only a few of them;

Point 4-1. What was the references number of the primary and secondary antibodies? What was the final concentration they were applied?

Response 4-1: Based on the reviewers' comments, we conducted a thorough review of our manuscript to address inaccuracies. Additionally, we added new sentences in the "Materials and Methods" section of the revised manuscript to provide information regarding the reference numbers and final concentrations of antibodies.

Old: Antibodies against human ERK, phospho-ERK, AKT, phospho-AKT, p38MAPK, phospho-p38MAPK, JNK, and phospho-JNK were acquired from Cell Signaling Technology (Danvers, MA, USA).

New: Antibodies against human ERK (#9102), phospho-ERK (#9106), AKT (#9272), phospho-AKT (#9271), p38MAPK (#9212), phospho-p38MAPK (#9215), JNK (#9252), and phospho-JNK (#4668) were acquired from Cell Signaling Technology (Danvers, MA, USA). The membrane was blocked with 5 % skim milk in Tris-buffered saline containing 0.1 % Tween-20 for 1 h at room temperature and probed with human α-tubulin (diluted 1:10,000), ICAM-1 (diluted 1:1,000), ERK (diluted 1:5,000), phospho-ERK (diluted 1:2,000), AKT (diluted 1:5,000), phospho-AKT (diluted 1:2,000), p38MAPK (diluted 1:5,000), phospho-p38MAPK (diluted 1:2,000), JNK (diluted 1:5,000) and phospho-JNK (diluted 1:2,000) antibodies for 16 h at 4 °C.

Point 4-2. How was the qPCR data evaluated, the mRNA level calculated? using the ∆∆Cq?

Response 4-2: We employed the Delta Delta Ct (∆∆Ct), also known as the Livak method, which is a widely used approach for analyzing qPCR data [Thomas D Schmittgen and Kenneth J Livak, “Analyzing real-time PCR data by the comparative CT method.” Nature Protocols, 3: 1101-1108, 2008]. We have provided a more detailed explanation of the qPCR analysis in the "Materials and Methods" section of the revised manuscript. Additionally, we have included new citations that support our methods in the "References" section of the revised manuscript.

Point 4-3. What percentage of SDS gel was used for protein fractionation

Response 4-3: In this study, we used 10% polyacrylamide gels for western blot analysis. Following your suggestion, we have included the polyacrylamide gel percentage in the “Materials and Methods” section of the revised manuscript.

* The original manuscript was thoroughly reviewed by a professional English editing service prior to submission. Moreover, our revised manuscript was also proofread by a professional English editing service before its resubmission. Please refer to the attached certificate for proof of English editing in “Non-published material”.

Round 2

Reviewer 2 Report

x